# Interaction Between Nitric Oxide and Silicon on Leghaemoglobin and S-Nitrosothiol Levels in Soybean Nodules

**DOI:** 10.3390/biom14111417

**Published:** 2024-11-07

**Authors:** Da-Sol Lee, Ashim Kumar Das, Nusrat Jahan Methela, Byung-Wook Yun

**Affiliations:** Department of Applied Biosciences, College of Agriculture and Life Sciences, Kyungpook National University, Daegu 41566, Republic of Korea; giftanna@naver.com (D.-S.L.); ashim@knu.ac.kr (A.K.D.); methela.ag@nstu.edu.bd (N.J.M.)

**Keywords:** ferric leghaemoglobin reductases, leghaemoglobin, nitric oxide, nodule formation, silicon, soybean plants

## Abstract

Nitrogen fixation in legume nodules is crucial for plant growth and development. Therefore, this study aims to investigate the effects of nitric oxide [S-nitrosoglutathione (GSNO)] and silicon [sodium metasilicate (Si)], both individually and in combination, on soybean growth, nodule formation, leghaemoglobin (Lb) synthesis, and potential post-translational modifications. At the V1 stage, soybean plants were treated for 2 weeks with 150 µM GSNO, and Si at concentrations of 1 mM, 2 mM, and 4 mM. The results showed that NO and Si enhance the nodulation process by increasing phenylalanine ammonia-lyase activity and Nod factors (*NIP2-1*), attracting rhizobia and accelerating nodule formation. This leads to a greater number and larger diameter of nodules. Individually, NO and Si support the synthesis of Lb and leghaemoglobin protein (*Lba*) expression, ferric leghaemoglobin reductases (*FLbRs*), and S-nitrosoglutathione reductase (*GSNOR*). However, when used in combination, NO and Si inhibit these processes, leading to elevated levels of S-nitrosothiols in the roots and nodules. This combined inhibition may potentially induce post-translational modifications in *FLbRs*, pivotal for the reduction of Lb^3+^ to Lb^2+^. These findings underscore the critical role of NO and Si in the nodulation process and provide insight into their combined effects on this essential plant function.

## 1. Introduction

Sustainable agricultural practices prioritise maintaining soil health, encompassing its physical, chemical, and biological properties to enhance nutrient uptake from roots to shoots [1,2,3]. However, despite the abundance of nutrients in the soil, many remain inaccessible to plants. For example, atmospheric dinitrogen (N_2_), which constitutes 78% (vol/vol) of the air in aerobic soil, is not readily available to plants [4]. The entire developmental process of plants, including protein biosynthesis, nucleic acids, chlorophyll, and certain vitamins [5], depends on nitrogen (N). Since soil often contains insufficient levels of N, plants have developed a complex mutualistic relationship with soil microbes [3,4,6]. Legumes, in particular, are renowned for their symbiotic relationship with microbes that form nodules, a process originally termed biological nitrogen fixation or symbiotic nitrogen fixation (BNF/SNF) [7,8].

Leguminous plants are classified into two main nodule types: indeterminate (elongated and found in species such as *Pisum sativum*, *Medicago sativa*, *Trifolium repens*, and *Vicia faba*) and determinate (spherical and occur in *Glycine max, Phaseolus vulgaris, and Vigna radiata*) [9,10]. The nodule is the site where nitrogenase catalyses the conversion of dinitrogen (N_2_) into two molecules of ammonia (NH_3_), which plants can absorb. However, nitrogenase is highly sensitive to oxygen (O_2_), which can irreversibly damage and inhibit its activity [7,11]. To regulate O_2_ levels, nodules contain ~40% leghaemoglobins (Lbs), which are proteins with a heme group capable of rapidly binding O_2_ and releasing it slowly. This mechanism helps to maintain free O_2_ concentrations of approximately 10 nM within the nodules, essential for bacterial survival, and supports efficient N fixation [11]. Fuchsman and Appleby [12] identified four primary types of Lbs in soybean nodules: *Lba*, *Lbc1*, *Lbc2*, and *Lbc3*, along with four minor variants—*Lbb*, *Lbd1*, *Lbd2*, and *Lbd3*—that are produced through N-terminal acetylation. As nodules mature, the ratio of *Lba* to *Lbc3* increases significantly. This shift suggests that *Lba* plays a crucial role in regulating O_2_ levels during the later stages of nodule development.

In contrast, understanding the complex interaction between Lbs and nitric oxide (NO), a small free radical molecule that regulates numerous plant cellular processes [13], is crucial within the nodule owing to their multifaceted roles. NO reacts rapidly with ferrous leghaemoglobin (Lb^2+^) to form nitrosyl-leghaemoglobin (Lb^2+^NO) [14]. This reaction typically occurs with the reduced form of ferric leghaemoglobin (Lb^3+^), facilitated by ferric leghaemoglobin reductase (FLbR) [15,16]. Moreover, the mechanism by which Lb^2+^ binds to O_2_ to regulate bacterial respiration is well-documented [17]. However, the inactivity of Lb^2+^NO in binding O_2_ remains unclear [10]. NO is known to inhibit nitrogenase activity [18,19,20], but its low levels can help balance nodule functions [21,22]. Additionally, NO is modulated through S-nitrosylation, a redox-based post-translational modification involving the addition of a NO moiety to a reactive cysteine thiol [23]. Maiti, et al. [24] confirmed S-nitrosylation in *Arachis hypogaea* nodule extracts, while approximately 80 S-nitrosylated proteins were identified in *Medicago truncatula*–*Sinorhizobium meliloti* mature nodules [25], although their functions have seldom been explored. Additionally, NO plays a role in various stages of the legume-rhizobia communication, including recognition, infection, nodule development, and nodule senescence [7,26,27,28].

Similarly, silicon (Si) is recognised as a conducive element for plant growth and development when supplied in the form of monosilicic acid [Si(OH)_4_] [29,30]. While some studies have demonstrated the effects of Si application on root nodulation [31,32,33,34], determining the optimal Si concentration for nodule formation and function remains challenging. However, transcriptomic analysis revealed that Si is involved in isoflavonoid synthesis, which promotes increased nodule formation [35]. Moreover, a recent preprint by Coquerel, et al. [36] demonstrates that Si application results in the accumulation of 989 and 212 differentially expressed proteins in the nodules of *Trifolium incarnatum* and its symbiotic bacteria (*Rhizobium leguminosarum* bv *trifolii*), respectively. These proteins are primarily involved in the synthesis of organic nitrogen compounds, amides, and peptides, which may explain the significant increase in N content in the nodules owing to Si application. Si influences Lb production in two pigeonpea cultivars [37]. However, the mechanism by which Si affects Lb synthesis remains unexplored.

These highlight the significant effect of NO and Si on the root nodulation process, though several questions remain unanswered. Therefore, this study aims to investigate the complex mechanisms of Lb synthesis and assess the effects of S-nitrosothiols (SNO) through the application of GSNO and Si in soybean plants. Additionally, the effects of GSNO and Si on soybean growth, chlorophyll content, phenylalanine ammonia-lyase (PAL) activity, and gene expression were investigated to explore the potential interactions between these molecules during the soybean root nodulation process. This study is the first to thoroughly demonstrate the role of NO and Si donors in the interaction between Lb production and SNO in soybean nodules.

## 2. Materials and Methods

### 2.1. Plant Materials, Growth Conditions, and Experimental Design

Pungsannamul, an elite soybean cultivar yielding 3–4 ton ha^−1^, was used in this study. The seeds, obtained from the Soybean Genetic Resource Center, Kyungpook National University, Daegu, Republic of Korea, were first surface-sterilised with 70% ethanol and washed five times with autoclaved distilled water (dH_2_O). Subsequently, the seeds (one seed/pipe) were sown in PVC pipes (diameter 7.5 cm × height 40 cm). The pipes were filled with 1.8 kg of sandy soil, and no additional fertiliser was applied. The study was conducted in the greenhouse at Kyungpook National University, where a 16-h light and 8-h dark photoperiod cycle was maintained while the temperature was kept at 28 ± 2 °C. The experimental design followed a complete randomised design conducted in triplicates.

At the V1 stage, soybean seedlings were divided into eight clusters: (i) control (water only), (ii) 150 µM GSNO [GSNO], (iii) 1 mM silicon [1Si], (iv) 2 mM silicon [2Si], (v) 4 mM silicon [4Si], (vi) 150 µM GSNO + 1 mM silicon [GSNO + 1Si], (vii) 150 µM GSNO + 2 mM silicon [GSNO + 2Si], and (viii) 150 µM GSNO + 4 mM silicon [GSNO + 4Si]. GSNO, a nitric oxide donor, was prepared by dissolving sodium nitrite (NaNO_2_; Sigma-Aldrich (St. Louis, MO, USA), CAS Number:7632-00-0) in dH_2_O to an equimolar concentration with reduced glutathione (GSH; Wako Special Grade, CAS Number: 70-18-8), which was dissolved in 1N hydrochloric acid (HCl). Additionally, sodium metasilicate (Na_2_SiO_3_; Sigma-Aldrich, CAS Number: 6834-92-0) was used as a silicon donor. The concentrations for both GSNO and Si were selected based on previous reports that demonstrated their optimal effects on plants [38,39,40,41]. Treatments were administered by applying 200 mL of dH_2_O, GSNO, Si, or their combinations through irrigation for 14 days (once daily) to the clusters mentioned above. On the day following the treatment period, shoots and roots were separated by cluster to assess seedling growth and nodule-related attributes. Furthermore, shoots, roots, and nodules were wrapped in aluminium foil, individually placed into liquid nitrogen, and stored in a −80 °C freezer for further analysis.

### 2.2. Chlorophyll Content Measurement

The methodology described by Hiscox and Israelstam [42] was used to determine the total chlorophyll content. For chlorophyll extraction, 100 mg of fresh soybean leaves were submerged in 20 mL of dimethyl sulphoxide (DMSO) and incubated at 60 °C for 4 h. After incubation, the samples were cooled to a temperature of 25 °C and transferred to a 96-well plate for absorbance measurement at 663 and 645 nm using a UV spectrophotometer (Thermo Scientific™ Multiskan™ GO Microplate Spectrophotometer, Ratastie, Finland). The total chlorophyll content was calculated in mg per gram of fresh weight using the following formula:

Total Chlorophyll = [(20.2 × Absorbance at 645) + (8.02 × Absorbance at 663)] × extract volume/weight in grams × 1000.

### 2.3. Estimation of Phenylalanine Ammonia-Lyase Activity

Phenylalanine ammonia-lyase (PAL) activity was assessed following the method described by Fan, et al. [43]. Briefly, 1 g of fresh soybean leaves was homogenised in an extraction buffer comprising 6.5 mL of Tris-HCl buffer (50 mM, pH 8.8) and 15 mM β-mercaptoethanol. Subsequently, the homogenate was centrifuged at 4000 rpm for 1 h, and the resulting supernatant was collected for the enzymatic assay. For the assay, a mixture containing 1 mL of the extraction buffer, 0.4 mL of dH_2_O, 0.5 mL of 10 mM L-phenylalanine, and 0.1 mL of the supernatant from each treatment was incubated at 37 °C for 1 h. The reaction was terminated by adding 0.5 mL of ethyl acetate, followed by centrifugation. The resulting precipitate was dissolved in 3 mL of 0.05 M NaOH, and the absorbance was measured at 290 nm using cinnamic acid standards. PAL activity was expressed as units per gram of protein.

### 2.4. Determination of Leghaemoglobin Content

Lb content was measured using the cyanmethemoglobin method described by Wilson and Reisenauer [44]. Briefly, roots and nodules were ground in liquid N. Subsequently, 100 mg of the pulverised material was homogenised in 0.6 mL of Drabkin’s solution (prepared by dissolving 52 mg KCN, 198 mg K_4_Fe(CN)_6_, and 1 g NaHCO_3_ in 1 L of dH_2_O). The homogenate was centrifuged at 10,000 rpm for 15 min at 4 °C. The supernatant was collected in a 2 mL e-tube, while the solid residue was re-homogenised with Drabkin’s solution and subjected to a second centrifugation. The supernatants from both centrifugations were combined, adjusted to a total volume of 2 mL with Drabkin’s solution, and centrifuged for an additional 30 min at 14,000 rpm at 4 °C. Subsequently, the clarified supernatant was used to measure absorbance at 540 nm against Drabkin’s solution in a 96-well plate using a UV spectrophotometer (Thermo Scientific™ Multiskan™ GO Microplate Spectrophotometer, Ratastie, Finland). A reference curve was generated using haemoglobin and five dilutions of bovine haemoglobin in the absence of sample extract. Results were expressed as milligrams of Lb per gram of fresh roots and nodules.

### 2.5. Measurement of S-Nitrosothiol

SNO concentrations were measured using a Sievers NOA-280i Nitric Oxide Analyser (Estero, FL, USA), following the method described by Yun, et al. [45]. First, 200 mg of fresh soybean shoots, roots and nodules were separately ground in liquid N and homogenised with 1X PBS buffer (pH~7.4). Subsequently, the samples were centrifuged at 14,000 rpm for 10 min at 4 °C, and the resulting supernatant was transferred to a new e-tube. The protein concentration of each sample was determined using the Bradford assay with a Coomassie (Bradford) protein assay kit (Thermo Fisher Scientific, Waltham, MA, USA), following the instructions of the manufacturer. Briefly, 1.5 mL of Coomassie dye reagent was added to 30 μL of the extracted protein in a 96-well plate and mixed thoroughly by pipetting. The absorbance of the samples was measured at 595 nm using a UV spectrophotometer (Thermo Scientific™ Multiskan™ GO Microplate Spectrophotometer, Ratastie, Finland). Subsequently, 100 μL of the extracted proteins were injected into the reaction vessel of the Nitric Oxide Analyser, which contained a CuCl/cysteine reducing agent, and the peak values were recorded. A CysNO-mediated standard curve was used to determine the SNO content. Finally, the SNO level was estimated as nM μg^−1^ of protein. Additionally, we employed GPS-SNO 1.0 software to predict and identify cysteine residues that may undergo NO-based modification through S-nitrosylation [46].

### 2.6. Relative Gene Expression Analysis

Quantitative real-time PCR (qRT-PCR) was conducted to examine the gene expression of S-nitrosoglutathione reductase 1 (*GSNOR1*), putative Si transporter (*NIP2-1*), also known as NOD26-like intrinsic protein 4-1, leghaemoglobin protein (*Lba*), and leghaemoglobin reductase genes (*FLbR-1* and *FLbR-2*). Samples were ground in liquid N, and RNA was extracted using the Biofact™ Total RNA Prep Kit. cDNA synthesis was performed using the Solg™ RT-kit according to the instructions of the manufacturer. Subsequently, the synthesised cDNA was used as a template to evaluate relative gene expression using the Real-Time PCR system (CFX Duet, Bio-RAD, Hercules, CA, USA) with the Solg™ 2× Multiplex Real-Time PCR Smart mix, which includes SYBR Green. Gene expression assays were conducted in triplicate. Appendix A shows the primer sequences.

### 2.7. Statistical Analysis

Data were collected and analysed by calculating the means and standard deviations using a one-way analysis of variance (ANOVA) with a 5% significance level. Means indicating significant differences were separated using the Tukey’s HSD test with Statistix 10 (https://www.statistix.com/, accessed on 6 November 2024). Boxplots and bar graphs were generated with the ‘ggplot2’, ‘tidyverse’, ‘hrbrthemes’, and ‘viridis’ packages in RStudio [47,48]. Significant differences between means were indicated with letters. All experiments were conducted in triplicate.

## 3. Results

### 3.1. Effects of S-Nitrosoglutathione and Silicon Supplementation on Chlorophyll Content and Phenylalanine Ammonia-Lyase Activity

The application of GSNO and Si, individually or in combination, did not significantly alter the chlorophyll content in soybean plants. However, the chlorophyll content was consistently higher when GSNO and Si were applied separately (Figure 1A). Furthermore, the activity of the enzyme PAL, crucial for flavonoid synthesis, was evaluated in response to GSNO and Si supplementation. The results showed that GSNO and Si, whether applied individually or in combination, generally increased PAL activity in the roots, (except in the 2Si treatment) (Figure 1C). In the shoot, PAL activity only increased with GSNO application, while Si application decreased it compared to that of the Control (Figure 1B). Nevertheless, the concurrent application of GSNO and Si enhanced PAL activity in the shoot (by 40.97%, 29.13%, and 18.42%) and root (by 6.87%, 25.40%, and 17.33%) compared to treatments with 1Si, 2Si, and 4Si, respectively (Figure 1B,C).

### 3.2. Application of S-Nitrosoglutathione and Silicon Regulates Soybean Nodulation

Soybean plants treated with GSNO and Si, individually or in combination, for 2 weeks exhibited the following increases in the number and diameter of nodules compared to that of the Control: GSNO (45.31% and 28.00%), 1Si (12.50% and 4.81%), 2Si (18.75% and 13.59%), 4Si (9.38% and 10.44%), GSNO+1Si (40.63% and 30.86%), GSNO+2Si (43.75% and 20.24%), and GSNO+4Si (60.94% and 8.10%, respectively) (Figure 2A,B). Conversely, the total nodule fresh weight was reduced in soybean plants treated with the combined application of GSNO and Si compared to that of the Control plants; however, results remained comparable all the treatments (Figure 2C).

To assess the effect of GSNO and Si on the Nod factor, which specifically regulates root nodulation, the relative expression of the NOD26-like intrinsic protein 2-1/silicon transporter gene (*NIP2-1*) was measured in the roots and nodules. In the roots, Si treatment resulted in a significant reduction in *NIP2-1* expression, with decreases of 16.98%, 46.13%, and 65.88% observed at increasing Si levels (1Si, 2Si, and 4Si, respectively) compared to that of the Control plants (Figure 2D). While GSNO alone increased *NIP2-1* expression in roots compared to the Control, the combination of GSNO with Si resulted in reduced expression levels by 12.31%, 45.44%, and 17.68% in the GSNO+1Si, GSNO+2Si, and GSNO+4Si treatments, respectively, compared to that of GSNO alone. However, GSNO and Si, whether applied individually or in combination, significantly upregulated *NIP2-1* expression in the nodules (Figure 2E). The treatments GSNO+1Si, GSNO+2Si, and GSNO+4Si exhibited 1086.00%, 852.60%, and 321.28% higher *NIP2-1* expression in the nodules compared to that of the Control, while a decreasing trend was observed, as *NIP2-1* expression in GSNO+2Si and GSNO+4Si was 19.68% and 64.48% lower, respectively, than in GSNO+1Si (Figure 2E). These findings indicate that *NIP2-1* expression modulation is influenced by GSNO and Si levels and is tissue-specific.

### 3.3. Response of S-Nitrosoglutathione and Silicon to Leghaemoglobin Levels in Soybean Roots and Nodules

In soybean nodules, treatment with GSNO and Si, individually or in combination, significantly increased Lb levels compared to that of the Control (Figure 3A,B). However, when compared to GSNO alone, the combinations of GSNO and Si reduced Lb content by 23.41%, 11.62%, and 19.30% in the GSNO+1Si, GSNO+2Si, and GSNO+4Si treatments, respectively (Figure 3B). Similar reductions in Lb content were observed when compared to treatments with single Si alone. In roots, GSNO and Si had a minimal effect on Lb content when applied individually, despite higher levels when compared to that of the Control (Figure 3F). Furthermore, the pattern of Lb content reduction in roots with combined GSNO and Si closely mirrored that observed in nodules, compared to treatments with GNSO or Si alone (Figure 3F).

*Lba* expression in soybean nodules increased significantly with GSNO and Si treatments compared to that of the Control. However, in the GSNO+1Si, GSNO+2Si, and GSNO+4Si treatments, *Lba* expression decreased by 67.55%, 69.57%, and 77.03%, respectively, compared to that of GSNO alone (Figure 3C). Similar reductions in *Lba* expression were observed in nodules with single Si treatments and GSNO combinations. In roots, GSNO exhibited a negative regulatory effect on *Lba* expression, similar to the effects of 2Si and 4Si treatments, while 1Si and GSNO+1Si significantly increased *Lba* expression by 58.21% and 264.86%, respectively, compared to that of control roots (Figure 3G). Additionally, the combination of GSNO and Si resulted in an increase in *Lba* expression compared to that of single treatments, contrasting with the nodular *Lba* expression patterns.

Compared to the Control, the expression of two Lb^3+^ genes (*FLbR-1* and *FLbR-2*) significantly increased in soybean nodules when treated individually with GSNO or Si or in combination, except for the GSNO+4Si treatment (Figure 3D,E). However, reductions of 58.62%, 19.23%, and 62.07% in *FLbR-1* expression and 58.03%, 14.72%, and 52.89% in *FLbR-2* expression were observed in the combined treatments of GSNO+1Si, GSNO+2Si, and GSNO+4Si, respectively, compared to the treatments with 1Si, 2Si, and 4Si (Figure 3D,E). Moreover, the expression of *FLbR-1* and *FLbR-2* in the roots exhibited a similar reduction pattern compared to the roots treated with 1Si, 2Si, and 4Si (Figure 3H,I). However, the application of GSNO and 1Si resulted in increased *FLbR-1* expression by 18.09% and 45.03%, respectively, and in *FLbR-2* expression by 36.62% and 80.25%, respectively, compared to the roots of the Control plants (Figure 3H,I). Increased Si levels alone led to a significant reduction in the expression of *FLbR-1* and *FLbR-2* in the roots and nodules.

### 3.4. Effect of S-Nitrosoglutathione and Silicon on S-Nitrosothiol Levels

Compared to the Control plants, GSNO significantly increased SNO levels by 246.52% in the root, 711.71% in the nodule, and 61.32% in the shoot (Figure 4A–C). Additionally, while Si treatments (1Si, 1Si, and 4Si) increased SNO levels in the shoot, root, and nodule, these increases were significantly lower than those induced by GSNO alone. Exceptions included the shoot treated with 1Si (exhibiting a 100.00% increase) and the nodule treated with 2Si (exhibiting a 229.50% increase), compared to that of the Control (Figure 4A–C). When the GSNO and Si were combined, a significant increase in SNO levels was observed in the shoot, root, and nodule. Specifically, SNO levels increased in the shoot (by 28.21%, 89.30%, and 41.83%), root (by 42.97%, 64.11%, and 52.75%), and nodule (by 36.27%, 29.64%, and 93.90%), compared to treatments with 1Si, 2Si, and 4Si, respectively (Figure 4C).

In contrast, combining GSNO with Si treatments (GSNO+1Si, GSNO+2Si, and GSNO+4Si) resulted in reductions in the relative expression of *GSNOR* compared to treatments with 1Si, 2Si, and 4Si. GSNOR expression specifically decreased by 18.81%, 23.14%, and 19.01% in the shoot; by 36.55%, 0.82%, and 60.14% in the root; and by 28.09%, 29.72%, and 22.26% in the nodule, respectively (Figure 4D–F). Compared to the GSNO treatment alone, these reductions were more pronounced, with decreases of 67.58%, 62.28%, and 41.40% in the shoot; 28.18%, 48.35%, and 75.07% in the root; and 81.47%, 66.25%, and 62.18% in the nodule for GSNO+1Si, GSNO+2Si, and GSNO+4Si, respectively (Figure 4D–F).

### 3.5. Effect of S-Nitrosoglutathione and Silicon on Soybean Shoot and Root Growth

Following a 14-day application of GSNO and Si, shoot length improved when these treatments were applied individually. However, their combined application resulted in the arrest of shoot length (Figure 5A,B). The combined treatments of GSNO+1Si, GSNO+2Si, and GSNO+4Si led to significant improvements in shoot fresh weight, with increases of 27.25%, 31.55%, and 24.71%, respectively, compared to that of GSNO alone. These treatments enhanced shoot fresh weight by 22.58%, 36.12%, and 17.41% compared to 1Si, 2Si, and 4Si, respectively (Figure 5D). A consistent trend was observed in root fresh weight with the application of GSNO and Si (Figure 5E), although they did not respond significantly. In contrast, GSNO and Si, either separately or combined, enhanced root length (Figure 5C).

## 4. Discussion

Global soybean production exceeded 395.91 million metric tonnes in the 2023/2024 period [49]. Soybean plants demonstrate their capacity to fix N in the soil through nodules, contributing an estimated 70% of the total 35.5 teragrams (Tg) of N fixed globally in 2018 [50]. Nodules, formed through the legume–rhizobia association, are a major source of N production in soybean plants and may be influenced by various external factors. Excessive N application promotes nodule senescence, thereby inhibiting BNF/SNF by decreasing Lb concentration and nitrogenase activity [51]. Additionally, studies show that NO inhibits nitrogenase activity and reacts with Lb^3+^ to form Lb^2+^NO, thought inactive in binding O_2_, thereby disrupting the fixation process. Moreover, Si is recognised for its role in modulating nodule formation. However, many of their (NO and Si) functions remain largely unexplored, limiting our understanding of the mechanisms controlling an effective nodulation system. Due to these reasons, this study focused on the impact of NO and Si on the rhizobia–legume symbiosis and Lb synthesis, which had previously been overlooked.

During the nodulation process, legume roots secrete flavonoid compounds recognised by rhizobia [52]. This recognition stimulates the rhizobia to release lipochitooligosaccharides, known as Nod factors [53], thereby initiating the legume–rhizobium association. Additionally, Nod factors are detected by LysM-type NF receptors (NFRs) in legumes, initiating rhizobial infection and nodule formation [54,55,56]. In this study, PAL activity, known for catalysing the conversion of phenylalanine in flavonoid biosynthesis [57,58], was assessed following a 14-day treatment with GSNO and Si, both individually and in combination. The results indicate that treatment with both elements increased PAL activity in the roots (Figure 1B,C), suggesting that soybean roots secrete more flavonoid compounds to attract rhizobia. Furthermore, the up-regulation of the Nod factor gene *NIP2-1* in the roots and nodules, along with the highest total number of nodules and largest nodule diameters, were observed with the addition of GSNO and Si (Figure 2A,B,D,E and Figure 6). Studies show that Si enhances total flavonoid levels in *M. truncatula*, nodule formation, and root length in legumes [31,34]. In contrast, the application of cPTIO, a NO inhibitor, reduces nodule formation and inhibits lateral root development [59]. Therefore, NO and Si are confirmed to play specific roles in enhancing the nodulation process. However, nodule fresh weight was inhibited when GSNO and Si were applied in combination, whereas individual applications did not produce this effect (Figure 2C). The reduction of nodule fresh weight suggests a different role of NO in BNF/SNF that was also reported previously [60].

The Lb within the nodule underscores its essential role in O_2_ binding, maintaining the survivability of bacterial symbionts necessary for effective N fixation in legumes. The red pigment found in soybean nodules was first reported by Kubo [61] and had not been previously detected in the roots, stems, or leaves [62]. However, recent studies have identified the presence of *VfLb29* in roots and leaves [63], and *GmLb5* in flowers [51]. While the individual application of GSNO and Si increased Lb content and *Lba* gene expression in nodules in this study, their combined application unexpectedly inhibited both (Figure 3A–C). The combination of 150 µM GSNO and 4 mM Si was particularly detrimental to Lb synthesis. This study reports, for the first time, on the detrimental effect of a NO donor when combined with Si. Additionally, previous studies demonstrate that nitrate supply alters the abundance of LbI and LbII [64]. Nitrite application forms LbNO, which is more challenging to dissociate into LbO_2_, the only form capable of supplying O_2_ to the symbionts [65]. However, while Lb content in soybean roots remained negligible, similar detrimental effects were observed in the nodules (Figure 3F). The combination of GSNO and Si led to increased *Lba* expression in the roots. However, despite this increased *Lba* gene expression, Lb content was reduced, which may be attributed to transcription factors such as *GmNAC039* and *GmNAC018* inducing early nodule senescence [66]. The overexpression of these factors decreases Lb levels and nitrogenase activity despite increased *Lba* expression.

The enzymatic reduction of Lb^3+^ to Lb^2+^ in legume nodules [67] is mediated by a protein known as FLbR, purified from soybean root nodules [68,69]. This 110 kDa homodimer, containing flavin adenine dinucleotide and utilising NAD(P)H as an electron donor, requires micromolar levels of O_2_ for the reduction process. The enzyme exhibits *K_m_* values of 7 mM for NADH and 9.5 mM for Lb^3+^, with a *V_max_* of 499 nmol Lb^2+^O_2_ formed per min per mg [69]. Given the critical role of this protein in N fixation, the current findings revealed a novel aspect: while GSNO and Si individually enhanced the expression of *FLbR-1* and *FLbR-2* genes in soybean roots and nodules, their combined application reduced the expression of the genes (Figure 3D,E,H,I and Figure 6). The ability of Lb to switch between the ferrous and ferric states allows dynamic regulation of O_2_ levels in the root nodules. This regulation is crucial for protecting the nitrogenase enzyme from O_2_-induced inactivation while ensuring adequate O_2_ availability for cellular respiration. However, while a concentration of 150 µM GSNO and 1 mM Si individually enhances the expression of these genes, higher levels of Si proved to be detrimental. Therefore, the complex interaction between NO and Si underscores the intricate mechanisms involved in Lb synthesis and its effect on the overall N fixation process in leguminous plants.

These findings reveal a connection between SNO levels and Lb synthesis. While the combined application of GSNO and Si led to reduced Lb content, *Lba* expression, and *FLbRs* gene expression (except for *Lba* in the roots) (Figure 3A–I), SNO levels exhibited an opposing trend (Figure 4A–C). Increased SNO levels enhance S-nitrosylation, a post-translational modification where NO attaches to cysteine residues in proteins [23,70]. This modification influences protein function, interactions, and stability, and is vital for signal transduction, enzyme regulation, and oxidative stress responses. Three potential S-nitrosylation sites were identified using GPS-SNO 1.0, indicating possible sites of S-nitrosylation (Appendix A). Furthermore, the addition of NO into the antioxidant tripeptide glutathione results in the formation of GSNO, being used as a mobile reservoir of NO activity. An enzyme known as GSNO reductase (GSNOR) specifically targets GSNO and is responsible for its metabolism, thereby regulating cellular levels of SNO [71]. Our findings suggest that the combined application of GSNO and Si reduced *GSNOR* expression in shoots, roots, and nodules more than the individual application of GSNO or Si (Figure 4D–F). Consequently, the findings suggest that the combination of GSNO and Si may regulate post-translational modifications through S-nitrosylation, leading to the inhibition of Lb synthesis. However, this hypothesis requires further investigation. Supplementation with GSNO and Si enhanced shoot and root fresh weight while reducing shoot length when combined (Figure 5A–E). Despite the shorter shoot length, their fresh weight increased significantly (Figure 5A,B,D), potentially owing to increased stem thickness.

## 5. Conclusions

In a nutshell, the aforementioned findings suggest that nodule formation was augmented by increasing the PAL activity and expression of Nod factors which is fine-tuned by NO and Si. Both NO and Si individually supported Lb synthesis and related proteins, crucial for nitrogen fixation; however, in combination they inhibited these processes, possibly triggering post-translational modifications in *FLbRs* suggesting complex interplays between NO and SI. Future research should focus on identifying the specific S-Nitrosylation sites as a post-translational modification. This will provide a deeper understanding of the roles of NO and Si in regulating nodule formation and function. Such studies could reveal the precise molecular interactions and pathways affected by NO and Si, offering insights into the mechanism underlying the contribution of these elements to soybean growth and development through their effect on the nodulation process.

## Figures and Tables

**Figure 1 biomolecules-14-01417-f001:**
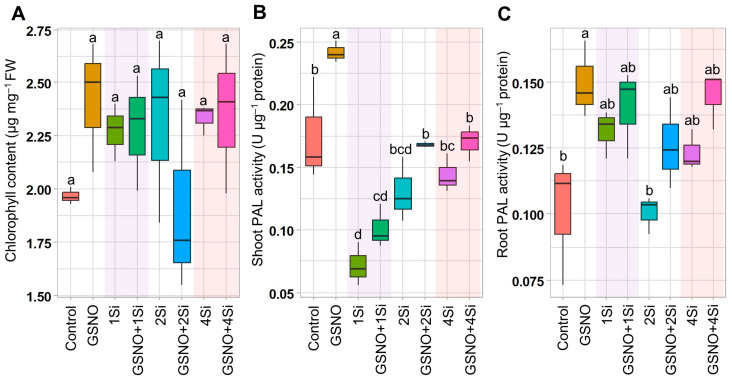
Analysis of chlorophyll content and PAL activity in soybean plants treated with GSNO (150 µM) and Si (1, 2, and 4 mM), both individually and in combination, for 2 weeks. The figure displays (**A**) chlorophyll content, (**B**) PAL activity in shoots, and (**C**) PAL activity in roots. Data are presented as means with standard errors (*n* = 3). Statistically significant differences between treatments, determined using the Tukey’s HSD test (*p* < 0.05), are indicated with different letters above the boxplots. Treatments include: dH_2_O-treated plants (Control), 150 µM GSNO-treated plants (GSNO), 1 mM Si-treated plants (1Si), 150 µM GSNO + 1 mM Si-treated plants (GSNO+1Si), 2 mM Si-treated plants (2Si), 150 µM GSNO + 2 mM Si-treated plants (GSNO+2Si), 4 mM Si-treated plants (4Si), and 150 µM GSNO + 4 mM Si-treated plants (GSNO+4Si).

**Figure 2 biomolecules-14-01417-f002:**
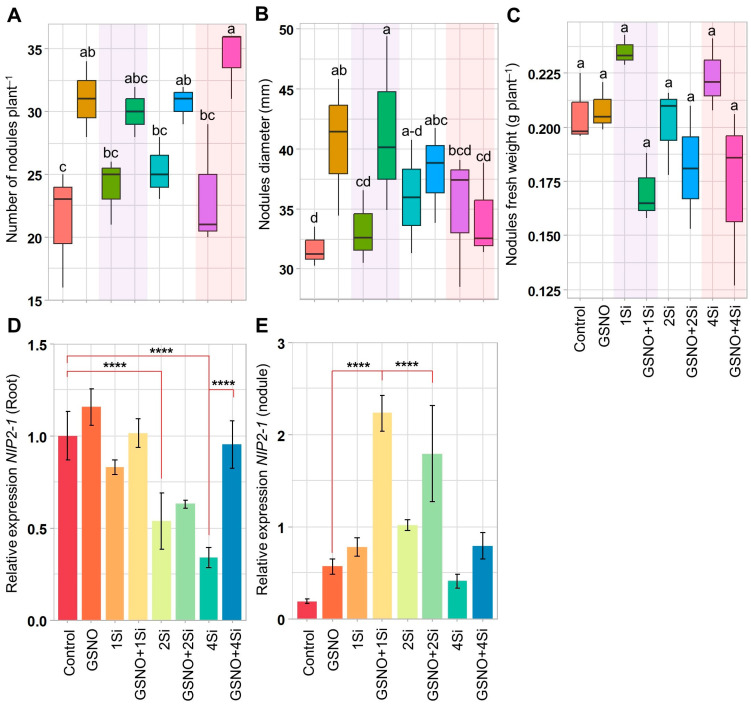
Effects of GSNO (150 µM) and Si (1, 2, and 4 mM), both individually and in combination, on nodule attributes in soybean plants after 2 weeks of treatment. The figure displays (**A**) the number of nodules per plant, (**B**) nodule diameter, (**C**) nodule fresh weight per plant, and (**D**) the relative gene expression of *NIP2-1* in roots and (**E**) nodules. Data are presented as means with standard errors (*n* = 3), except for nodule diameter (*n* = 10). For panels (**A**–**C**), statistical significance between treatments was assessed using the Tukey’s HSD test (*p* < 0.05), with different letters above the boxplots indicating significant differences. For panels (**D**,**E**), significant differences were determined using the Student’s *t*-test (**** *p* ≤ 0.0001). Treatments include: dH_2_O-treated plants (Control), 150 µM GSNO-treated plants (GSNO), 1 mM Si-treated plants (1Si), 150 µM GSNO + 1 mM Si-treated plants (GSNO+1Si), 2 mM Si-treated plants (2Si), 150 µM GSNO + 2 mM Si-treated plants (GSNO+2Si), 4 mM Si-treated plants (4Si), and 150 µM GSNO + 4 mM Si-treated plants (GSNO+4Si).

**Figure 3 biomolecules-14-01417-f003:**
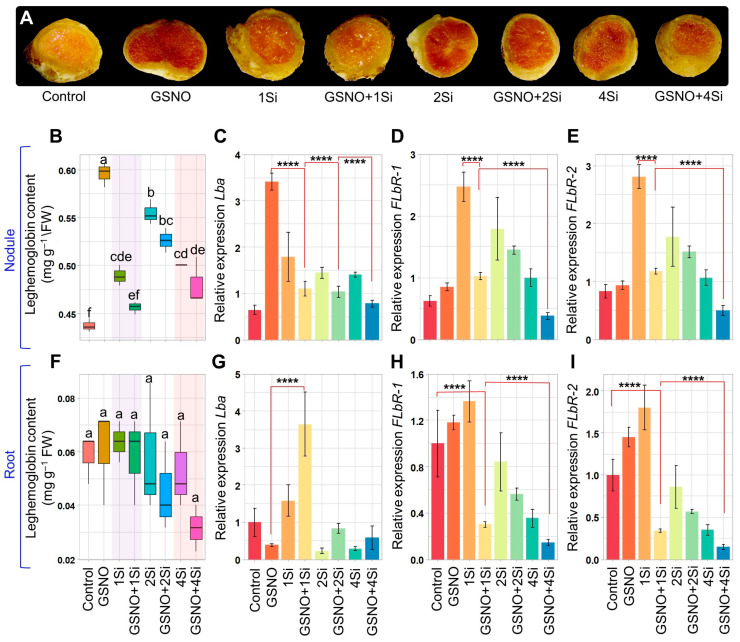
Assessment of leghaemoglobin content and gene expression in soybean nodules and roots treated with GSNO (150 µM) and Si (1, 2, and 4 mM) for 2 weeks, both individually and in combination. The figure illustrates (**A**) the cross section of nodules per treatment, (**B**,**F**) leghaemoglobin content in nodules and roots and (**C**,**G**) relative expression of *Lba* in nodules and in roots, (**D**,**H**) relative expression of *FLbR-1* in nodules and roots, and (**E**,**I**) relative gene expression of FLbR-2 in nodules and roots. Data are presented as means with standard errors (*n* = 3). For panels (**B**,**F**), statistically significant differences between treatments, determined using the Tukey’s HSD test (*p* < 0.05), are indicated with different letters above the boxplots. For panels (**C**–**E**,**G**–**I**), significant differences were assessed using the Student’s *t*-test (**** *p* ≤ 0.0001). Treatments include: dH_2_O-treated plants (Control), 150 µM GSNO-treated plants (GSNO), 1 mM Si-treated plants (1Si), 150 µM GSNO + 1 mM Si-treated plants (GSNO+1Si), 2 mM Si-treated plants (2Si), 150 µM GSNO + 2 mM Si-treated plants (GSNO+2Si), 4 mM Si-treated plants (4Si), and 150 µM GSNO + 4 mM Si-treated plants (GSNO+4Si).

**Figure 4 biomolecules-14-01417-f004:**
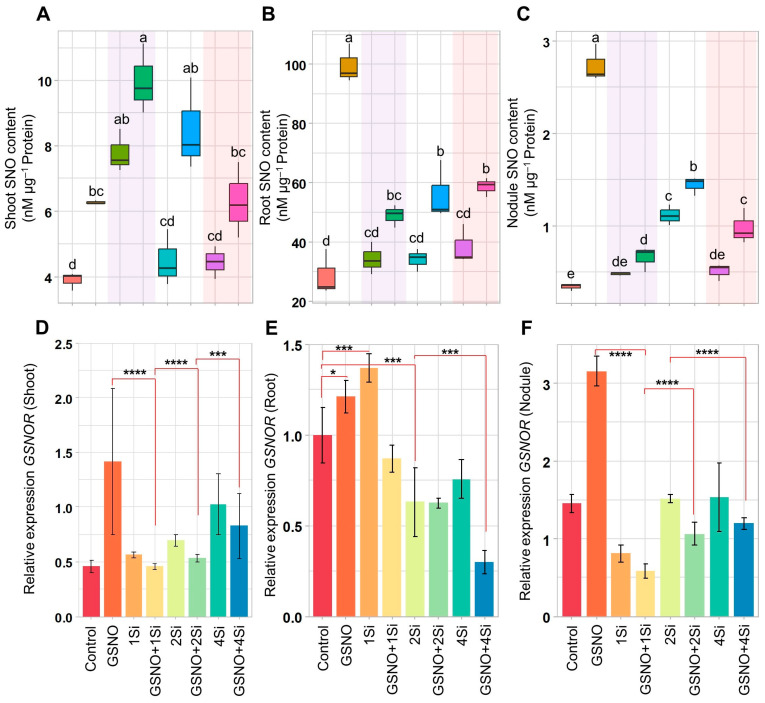
Measurement of S-nitrosothiols (SNO) levels and related gene expression in soybean plants treated with GSNO (150 µM) and Si (1, 2, and 4 mM) for 2 weeks, both individually and in combination. The figure presents (**A**–**C**) SNO level in the shoot, roots, and nodules and (**D**–**F**) relative expression of *GSNOR* in the shoot, roots, and nodules. Data are presented as means with standard errors (*n* = 3). For panels (**A**–**C**), statistically significant differences between treatments, determined using the Tukey’s HSD test (*p* < 0.05), are indicated with different letters above the boxplots. For panels (**D**–**F**), significant differences were assessed using the Student’s *t*-test (* *p* ≤ 0.05, *** *p* ≤ 0.001, and **** *p* ≤ 0.0001). Treatments include: dH_2_O-treated plants (Control), 150 µM GSNO-treated plants (GSNO), 1 mM Si-treated plants (1Si), 150 µM GSNO + 1 mM Si-treated plants (GSNO+1Si), 2 mM Si-treated plants (2Si), 150 µM GSNO + 2 mM Si-treated plants (GSNO+2Si), 4 mM Si-treated plants (4Si), and 150 µM GSNO + 4 mM Si-treated plants (GSNO+4Si).

**Figure 5 biomolecules-14-01417-f005:**
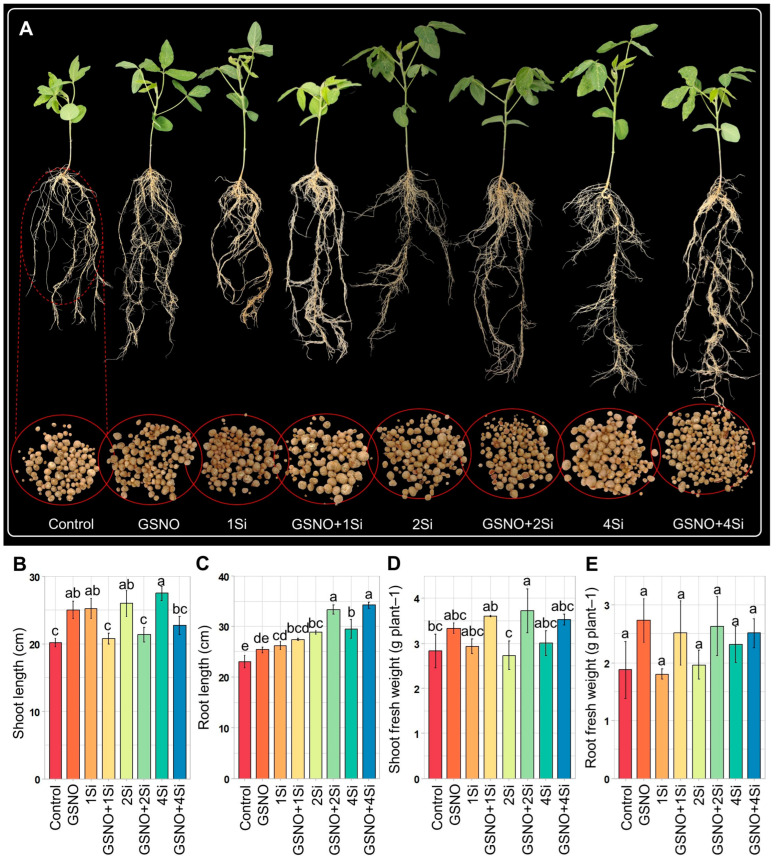
Evaluation of growth attributes of soybean plants treated with GSNO (150 µM) and Si (1, 2, and 4 mM), both individually and in combination, for 2 weeks. The figure presents: (**A**) The shoot, root, and nodule growth of soybean plants treated for 2 weeks. (**B**) Growth parameters include shoot length, (**C**) root length, (**D**) shoot fresh weight, and (**E**) root fresh weight. Data are presented as means with standard errors (*n* = 3). Statistically significant differences between treatments, determined using the Tukey’s HSD test (*p* < 0.05), are indicated with different letters above the boxplots. Treatments include: dH_2_O-treated plants (Control), 150 µM GSNO-treated plants (GSNO), 1 mM Si-treated plants (1Si), 150 µM GSNO + 1 mM Si-treated plants (GSNO+1Si), 2 mM Si-treated plants (2Si), 150 µM GSNO + 2 mM Si-treated plants (GSNO+2Si), 4 mM Si-treated plants (4Si), and 150 µM GSNO + 4 mM Si-treated plants (GSNO+4Si).

**Figure 6 biomolecules-14-01417-f006:**
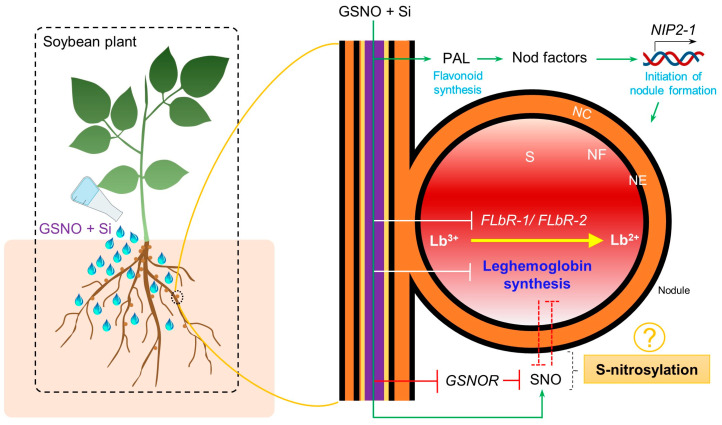
Effect of GSNO and Si on soybean nodulation processes, applied individually or in combination. While GSNO and Si separately enhanced leghaemoglobin synthesis, their combined application inhibited the synthesis and the expression of ferric leghaemoglobin reductases (*FLbR-1* and *FLbR-2*) genes, which are crucial for reducing Lb^3+^ to Lb^2+^ to maintain O_2_ homeostasis in the nodule. However, the combined application of GSNO and Si increased S-nitrosothiol (SNO) levels by decreasing the S-nitrosoglutathione reductase (*GSNOR*) gene. These findings suggest that S-nitrosylation occurs within the nodule via post-translational modification. Despite this, both individual and combined applications of GSNO and Si enhanced flavonoid synthesis (phenylalanine ammonia-lyase, PAL), attracting rhizobia with the aid of Nod factors, leading to increased nodule formation and N fixation. Solid lines indicate activation or inhibition, where dotted lines are hypothetical. NC, nodule cortex; NE, nodule endodermis; NF, nitrogen fixation zone; S, senescent zone.

## Data Availability

The data presented in this study are available on request from the corresponding author. The data are not publicly available due to privacy.

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
