# Peer review of "Interaction Between Nitric Oxide and Silicon on Leghaemoglobin and S-Nitrosothiol Levels in Soybean Nodules"

_biomolecules, 2024, doi:10.3390/biom14111417_

Round 1

Reviewer 1 Report

Comments and Suggestions for Authors

Comment 1: How is the concentration of GSNO determined? This needs to be explained.

Comment 2: In the Materials and Methods section, the concentration of GSNO is specified as 150 µM. However, in Line 15 of the Abstract, it is written as 200 µ M. Why? This mistake is very serious and requires clarification.

Comment 3: Is the use of a boxplot with only three repeats statistically valid and appropriate for the data presented?

Comment 4: In section 3.2, “Application of S-nitrosoglutathione and silicon regulates soybean nodulation”, it is observed that the combined application of GSNO and silicon enhances both the number and diameter of nodules. However, there is a significant decrease in the total nodule weight compared to the control. Could you explain this phenomenon? Additionally, please confirm whether the nodule weight depicted in Figure 2 represents fresh or dry weight.

Comment 5: The red lines indicating significant differences in Figures 3 and 4 could be improved to match the presentation style seen in Figure 2.

Comment 6: In Line 134, it should be 1Si, 2Si, and 4Si.

Comment 7: In Line 349, it should be Si (Fig. 5E). Figure 5 lacks a subgraph labeled E, and a similar issue is noted in Line 459.

Comment 8: In Line 459, it is imperative to specify whether the weight refers to fresh or dry weight to avoid any ambiguity.

Comment 9: In Line 460, stem thickness (Fig. 5D)? Similar to Question 7, Figure 5 needs to be modified and supplemented, or correct the manuscript section to align with the figure provided.

Author Response

Responses to the Reviewers’ Comments

Reviewer 1

Thank you very much for your overall comments of our manuscript. We also very much appreciate your constructive comments and suggestions that have helped us improve the quality of our manuscript.

Comment 1

How is the concentration of GSNO determined? This needs to be explained.

Response

Thank you very much for your insightful comments.

Nitric oxide (NO), a short-lived redox molecule, contributes to numerous physiological processes in plants, primarily through post-translational modifications such as S-Nitrosylation. The homeostasis of NO is tightly regulated by its biosynthesis and turnover, which operate through distinct mechanisms in animal and plant cells.

Before beginning our experiment, we reviewed the concentrations of different NO donors used in previous studies on plant growth and development. We found that most studies reported significant effects on soybean plants at concentrations between 100–200 µM (Vital et al. 2019; Khan et al. 2019; Ahmad et al. 2021; Basit et al. 2023). Furthermore, previous research from our lab has highlighted the effectiveness of NO donors within this range (Methela et al. 2023a; Methela et al. 2023b). Based on these findings, we selected a concentration of 150 µM to examine its impact on soybean nodules. (L. 113-114)

Moreover, we made correction in the revised file as well.

  1. Vital, R. G., Müller, C., da Silva, F. B., Batista, P. F., Merchant, A., Fuentes, D., ... & Costa, A. C. (2019). Nitric oxide increases the physiological and biochemical stability of soybean plants under high temperature. Agronomy9(8), 412.
  2. Khan, M. A., Khan, A. L., Imran, Q. M., Asaf, S., Lee, S. U., Yun, B. W., ... & Lee, I. J. (2019). Exogenous application of nitric oxide donors regulates short-term flooding stress in soybean. PeerJ7, e7741.
  3. Ahmad, P., Alyemeni, M. N., Wijaya, L., Ahanger, M. A., Ashraf, M., Alam, P., ... & Rinklebe, J. (2021). Nitric oxide donor, sodium nitroprusside, mitigates mercury toxicity in different cultivars of soybean. Journal of Hazardous Materials408, 124852.
  4. Basit, F., Bhat, J. A., Alyemeni, M. N., Shah, T., & Ahmad, P. (2023). Nitric oxide mitigates vanadium toxicity in soybean (Glycine max L.) by modulating reactive oxygen species (ROS) and antioxidant system. Journal of Hazardous Materials451, 131085.
  5. Methela, N. J., Islam, M. S., Lee, D. S., Yun, B. W., & Mun, B. G. (2023a). S-nitrosoglutathione (GSNO)-mediated lead detoxification in soybean through the regulation of ROS and metal-related transcripts. International Journal of Molecular Sciences24(12), 9901.
  6. Methela, N. J., Pande, A., Islam, M. S., Rahim, W., Hussain, A., Lee, D. S., ... & Yun, B. W. (2023b). Chitosan-GSNO nanoparticles: a positive modulator of drought stress tolerance in soybean. BMC Plant Biology23(1), 639.

Comment 2

In the Materials and Methods section, the concentration of GSNO is specified as 150 μM. However, in Line 15 of the Abstract, it is written as 200 μ M. Why? This mistake is very

serious and requires clarification.

Response

We apologies for the mistake. We have corrected in the revised manuscript.

Comment 3

Is the use of a boxplot with only three repeats statistically valid and appropriate for the data presented?

Response

Thank you very much for raising this question. As you may know, a box plot typically requires a minimum of three individual replications, while a bar plot can be generated with even a single replication, depending on the software used. Although we agree that additional replications could enhance the representation, it is still acceptable to create a box plot with three replications. In fact, we have found several published articles in plant science (even in Biomolecules journal) that use box plots with three replications (Khamesa-Israelov et al. 2024; Wang et al. 2024; Azeem et al. 2023; Song et al. 2021).

Given that we constructed our box plot based on three replications, we believe it provides clear and understandable insight into the differences between treatment combinations. We would be very grateful if the reviewer could take this into consideration as we move forward with the next steps.

  1. Khamesa-Israelov, H., Finkelstein, A., Shani, E., & Chamovitz, D. A. (2024). Investigation of the Roles of Phosphatidylinositol 4-Phosphate 5-Kinases 7, 9 and Wall-Associated Kinases 1–3 in Responses to Indole-3-Carbinol and Biotic Stress in Arabidopsis Thaliana. Biomolecules, 14(10), 1253.
  2. Wang, Y., Zhang, L., Feng, L., Fan, Z., Deng, Y., & Feng, T. (2024). Influence of Functional Traits of Dominant Species of Different Life Forms and Plant Communities on Ecological Stoichiometric Traits in Karst Landscapes. Plants13(17), 2407.
  3. Azeem, A., Ul-Allah, S., Azeem, F., Naeem, M., Sattar, A., Ijaz, M., & Sher, A. (2023). Effect of foliar applied zinc sulphate on phenotypic variability, association and heritability of yield and zinc biofortification related traits of wheat genotypes. Heliyon9(9).
  4. Song, H., Chen, D., Sun, S., Li, J., Tu, M., Xu, Z., ... & Jiang, G. (2021). Peach-Morchella intercropping mode affects soil properties and fungal composition. PeerJ9, e11705.

Comment 4

In section 3.2, “Application of S-nitrosoglutathione and silicon regulates soybean nodulation”, it is observed that the combined application of GSNO and silicon enhances both the number and diameter of nodules. However, there is a significant decrease in the total nodule weight compared to the control. Could you explain this phenomenon? Additionally, please confirm whether the nodule weight depicted in Figure 2 represents fresh or dry weight.

Response

Thank you for your comment.

Nitric oxide production associated with nitrate supply could also mold the process of symbiotic nitrogen fixation (SNF)/ biological nitrogen fixation (BNF). It was reported that different levels of NO had different roles in SNF (Hichri et al., 2015). For example, the excessive uptake of nitrate led to the production of NO and further inhibited the activity of nitrogenase (Berger et al., 2019; Wang et al., 2020). Similarly, excessive N application promotes nodule senescence, thereby inhibiting BNF/SNF by decreasing leghaemoglobin concentration and nitrogenase activity (Du et al. 2020).

However, our findings indicate that the combined application of GSNO and Si enhanced flavonoid biosynthesis, which in turn stimulated the production of nod factors to initiate the nodule formation. Although we observed an increase in nodule number and weight, the occurrence of nitric oxide-induced nodule senescence ultimately led to a reduction in nodule fresh weight. Consequently, both our results and previous studies suggest that nitric oxide signaling may induce nodule formation through both dependent and independent pathways.

Moreover, we made correction in the revised file as well. (L. 397-398)

  1. Hichri, I., Boscari, A., Castella, C., Rovere, M., Puppo, A., & Brouquisse, R. (2015). Nitric oxide: a multifaceted regulator of the nitrogen-fixing symbiosis. Journal of experimental botany66(10), 2877-2887.
  2. Berger A, Boscari A, Frendo P, Brouquisse R. 2019. Nitric oxide signaling, metabolism and toxicity in nitrogen-fixing symbiosis. Journal of Experimental Botany 70: 4505–4520.
  3. Wang Q, Huang Y, Ren Z, Zhang X, Ren J, Su J, Zhang C, Tian J, Yu YJ, Gao GF et al. 2020. Transfer cells mediate nitrate uptake to control root nodule symbiosis. Nature Plants 6: 800–808.
  4. Du, M.; Gao, Z.; Li, X.; Liao, H. Excess nitrate induces nodule greening and reduces transcript and protein expression levels of soybean leghaemoglobins. Ann. Bot. 2020, 126, 61-72.

Comment 5

The red lines indicating significant differences in Figures 3 and 4 could be improved to match the presentation style seen in Figure 2.

Response

We have corrected in the revised manuscript.

Comment 6

In Line 114, it should be 1Si, 2Si, and 4Si.

Response

We have corrected in the revised manuscript.

Comment 7

In Line 349, it should be Si (Fig. 5E). Figure 5 lacks a subgraph labeled E, and a

similar issue is noted in Line 459.

Response

We apologies for the mistake. We have corrected in the revised manuscript.

Comment 8

In Line 459, it is imperative to specify whether the weight refers to fresh or dry weight to avoid any ambiguity.

Response

According to your suggestions, we have corrected in the revised manuscript.

Comment 9

In Line 460, stem thickness (Fig. 5D)? Similar to Question 7, Figure 5 needs to be modified and supplemented, or correct the manuscript section to align with the figure provided.

Response

We apologies for the mistake. We have corrected in the revised manuscript.

Reviewer 2 Report

Comments and Suggestions for Authors

The manuscript gives a good indication about the interaction between nitric oxide and silicon on leghaemoglobin and S-nitrosothiol levels in soybean nodules. 

Data were analyzed in terms of one way ANOVA, followed by LSD posthoc test. It is recommended first to check the normality of data whether parametric or nonparametric. 

On the other hand, it is also recommended to use a more conservative posthoc test e.g. Duncan's multiple range test (DMRTs) or Tukey's HSD.

Figure 1A the posthoc letters seems to be incorrect and need to be corrected descending from letter a, b , kindly revise Figure 1 A

Author Response

Responses to the Reviewers’ Comments

Reviewer 2

The manuscript gives a good indication about the interaction between nitric oxide and silicon on leghaemoglobin and S-nitrosothiol levels in soybean nodules.

Response

Thank you very much for your overall assessment of our manuscript.

Comment 1

Data were analyzed in terms of one-way ANOVA, followed by LSD posthoc test. It is recommended first to check the normality of data whether parametric or nonparametric.

Response

We are grateful to you for your great efforts to improve our manuscript. According to your query, we checked again our results which representing data are normally distributed (parametric), so to use one-way ANOVA with LSD test is the appropriate statistical analysis.

Comment 2

On the other hand, it is also recommended to use a more conservative posthoc test e.g. Duncan's multiple range test (DMRTs) or Tukey's HSD.

Response

We highly appreciate your comments and we also feel how you pushing us to enhance the quality of our manuscript.

We are agreed with your suggestions that DMRT or Tukey’s HSD can also give the statistical difference between the treatment combinations. However, so many published papers has been done with the LSD test when used one-way ANOVA to check the significant variations between the groups (Sultana et al. 2024; Das et al. 2022; Rahman et al. 2023).

Therefore, we believe that LSD also gives sufficient significant difference in our study. We would be very grateful if the reviewer could take this into consideration as we move forward with the next steps.

  1. Sultana, S., Rahman, M. M., Das, A. K., Haque, M. A., Rahman, M. A., Islam, S. M. N., ... & Mostofa, M. G. (2024). Role of salicylic acid in improving the yield of two mung bean genotypes under waterlogging stress through the modulation of antioxidant defense and osmoprotectant levels. Plant Physiology and Biochemistry206, 108230.
  2. Das, A. K., Anik, T. R., Rahman, M. M., Keya, S. S., Islam, M. R., Rahman, M. A., ... & Mostofa, M. G. (2022). Ethanol treatment enhances physiological and biochemical responses to mitigate saline toxicity in soybean. Plants11(3), 272.
  3. Rahman, M. M., Das, A. K., Sultana, S., Ghosh, P. K., Islam, M. R., Keya, S. S., ... & Mostofa, M. G. (2023). Biochar potentially enhances maize tolerance to arsenic toxicity by improving physiological and biochemical responses to excessive arsenate. Biochar5(1), 71.

Comment 3

Figure 1A the posthoc letters seems to be incorrect and need to be corrected descending from letter a, b , kindly revise Figure 1 A

Response

Thank you for your nice comment. We have checked and ensuring to you that our analysis and placement of the lettering are totally correct.

As box plot always shows the distribution of higher and lower value, it seems like that. Please see the median in the box plot then you can easily understand the difference. Therefore, we would be very grateful if the reviewer could take this into consideration as we move forward with the next steps.
